# Ursodeoxycholic acid inhibits feline infectious peritonitis virus infection through activating JAK-STAT signaling pathway-induced type I interferon

Yi Zhong,[1] Zhiwei Sun,[1] Ziyan Song,[1] Jinman Ding,[1] Yanwen Song,[1] Yi Li,[1] Guisong Liao,[1] Xin Wang,[1] Yan Zeng,[1] Nan Hu,[1] Xingcui Zhang,[1] Zhenhui Song[1]

**ABSTRACT** Feline infectious peritonitis virus (FIPV), a highly pathogenic subtype of feline coronavirus, is characterized by antigenic variability and immune evasion, resulting in limited efficacy of existing clinical therapeutic regimens. In recent years, ursodeoxycholic acid (UDCA), a hydrophilic bile acid derivative, has been found to have broad antiviral activity. Based on this, this study significantly improved the replication efficiency of FIPV by constructing a CRFK-PHBLV-N cell line that can stably express the FIPV N protein; UDCA had obvious antiviral effects, and signaling pathways related to UDCA's anti-FIPV activity were screened by transcriptome sequencing, and it was found that UDCA could significantly promote secretion of interferon-β by CRFK-PHBLV-N cells, activating the JAK-STAT signaling pathway and upregulating the expression of interferon-stimulated genes. This study revealed that FIPV escapes host immune response by inhibiting the JAK-STAT pathway, and UDCA was able to reverse this inhibition and enhance the expression of antiviral proteins, thus effectively inhibiting the replication and spread of FIPV. Furthermore, UDCA can also directly disrupt the envelope components of FIPV, inducing the disintegration of the viral structure. This study not only provides a new strategy for the efficient *in vitro* culture of FIPV, but also lays a theoretical foundation for the development of anti-FIPV drugs targeting the JAK-STAT pathway.

**IMPORTANCE** In this study, through the establishment of a complete research chain of "virus titer-enhanced cell model–drug effect evaluation–signaling pathway analysis", we confirmed that ursodeoxycholic acid (UDCA) can inhibit feline infectious peritonitis virus (FIPV) infection and revealed that UDCA can activate the JAK-STAT signaling pathway by promoting the production of large amounts of interferon-β in host cells to protect against FIPV infection. It not only provides a new experimental tool (CRFK-PHBLV-N cell line) and drug candidate (UDCA) for FIP prevention and control, but also lays a theoretical foundation for the development of novel therapeutic strategies targeting the host antiviral pathway through the elucidation of the JAK-STAT pathway mechanism.

**KEYWORDS** feline infectious peritonitis virus, lentivirus, stable cell line, ursodeoxycholic acid, JAK-STAT signaling pathway

Feline coronavirus (FCoV), a single-stranded, positive-stranded, and nonsegmented RNA virus, belongs to the family *Coronaviridae*, subfamily *Coronaviridae*, genus *Coronavirus* (1). It can be classified into two biological types according to its pathogenicity: feline enteric coronavirus (FECV) and feline infectious peritonitis virus (FIPV). FECV infection is essentially subclinical and confined mainly to the gastrointestinal tract and usually elicits only self-limiting symptoms such as mild diarrhea, transient fever, and appetite loss (2), and occasionally severe enterocolitis (3). FECV infection is usually

**Peer Reviewers** Tongling Shan, Shanghai Veterinary Research Institute, Chinese Academy of Agricultural Sciences, Shanghai, China; Bin Li, Jiangsu Academy of Agricultural Sciences, Institute of Veterinary Medicine, Nanjing, China

Address correspondence to Xingcui Zhang, zhangxc923@163.com, or Zhenhui Song, szh7678@126.com.

Yi Zhong and Zhiwei Sun contributed equally to this article. The author order was determined based on their contributions to the article.

Guisong Liao and Nan Hu contributed equally to this article.

Guisong Liao and Yan Zeng are joint senior authors.

The authors declare no conflict of interest.

See the funding table on p. 18.

characterized by subclinical symptoms, mainly confined to the gastrointestinal system, often causing only self-limiting symptoms such as mild diarrhea, transient fever, and loss of appetite, and occasionally severe enteritis (2). Unlike FECV, FIPV has a greater tissue tropism and can be disseminated systemically by infecting the monocyte-macrophage system, generating high titers of viral particles and causing systemic infection. FCoV has two serotypes: serotype I FCoV (serotype I FECV and serotype I FIPV) and serotype II FCoV (serotype II FECV and serotype II FIPV). The clinically prevalent strains of FCoV are mainly serotype I, but serotype I strains are difficult to isolate and culture *in vitro*, and the FCoV strains successfully isolated and cultured *in vitro* are mostly of serotype II. Two serotypes cause similar disease characteristics (3), and serotype II strains have become common models for laboratory research because they can stably replicate in passaged cell lines such as CRFK, FCWF-4 (4).

Ursodeoxycholic acid (UDCA) is a kind of bile acid derivative with a variety of pharmacological activities, which can not only effectively improve the symptoms of cholestasis but also has multiple mechanisms of action such as immunomodulatory, anti-inflammatory, and anti-apoptosis (5–7), showing promising prospects in the treatment of a variety of diseases. In recent years, it has been found that UDCA has a unique value in antiviral infections (8). It has been shown that UDCA can effectively inhibit SARS-CoV-2 infection by modulating the farnesoid X receptor (FXR)-ACE2 pathway, and this breakthrough finding provides a new idea for host-targeted therapy of coronavirus (9, 10), but the role of UDCA on FIPV is still unclear.

This study confirmed that UDCA can inhibit FIPV infection by establishing a complete research chain of "virus titer-enhancement cell model–pharmacological evaluation–signaling pathway analysis" and revealed that UDCA can activate the JAK-STAT signaling pathway by promoting the host cells to produce a large amount of interferon-β (IFN-β) and protect the host cells from FIPV infection. It not only provides a new experimental tool (CRFK-PHBLV-N cell line) and drug candidate (UDCA) for FIP prevention and control, but also lays a theoretical foundation for the development of novel therapeutic strategies targeting the host antiviral pathway through the elucidation of the JAK-STAT pathway mechanism.

## MATERIALS AND METHODS

### Antibodies and reagents

The mouse-derived Flag monoclonal antibody, rabbit-derived β-tubulin polyclonal antibody, fluorescent rabbit secondary antibody CY3, goat anti-rabbit IgG, and goat anti-mouse IgG secondary antibody were purchased from Proteintech (USA); mouse-derived β-tubulin polyclonal antibody was purchased from Sevier (Wuhan), and the rabbit polyclonal antibody against FIPV was prepared by the laboratory; DMSO was purchased from Bioengineering (Shanghai) Co., Ltd.; STAT1 rabbit polyclonal antibody, P-STAT1 rabbit polyclonal antibody, TLR3 rabbit polyclonal antibody, IRF3 rabbit polyclonal antibody, P-IRF3 rabbit polyclonal antibody, MX1 rabbit polyclonal antibody, and ISG15 rabbit polyclonal antibody were purchased from Wuhan Sanying. β-Tubulin and GAPDH were purchased from Wuhan Sevier. QuickCut *Xho* I, QuickCut *EcoR* I, and DNAMarker were purchased from Takara (Beijing). Ursodeoxycholic acid was selected for this study and was provided by MedChemExpress, a life science reagent service provider.

### Viruses, cell lines, and lentiviral vectors

CRFK cells, HEK-293T cells (cultured in DMEM medium), FIPV-CQ strain (GenBank: A22378.1), and lentiviral vectors (lentiviral core vector: PHBLV-3Flag-ZsGreen; lentiviral helper packaging plasmids: PSPAX2 and PMD2G) are kept in our laboratory.

## Viral titer assay

Viruses were proliferating and titrated on CRFK and CRFK-PHBLV-N cells, and viral titers were determined by the 50% tissue culture infectious dose (TCID50) assay using the Reed-Muench method and expressed as TCID50/mL.

## Enhancing FIPV virus titers using stable cell lines

By cloning the FIPV N gene into the lentiviral core plasmid and co-transfecting the auxiliary plasmids to the HEK-293T cells, the success of viral packaging was determined by the green fluorescence produced in the cells and by Western blot analysis. After confirmation, the successfully packaged lentiviral solution was used to infect CRFK host cells. Before infecting host cells, the optimal selection concentration of puromycin for the host cells was determined. The concentration at which all cells died after 8 days of culture in the selection medium was established as the optimal selection concentration for subsequent experiments, and a cell survival curve was plotted. The target cells infected with the lentivirus solution were added to the optimal selection concentration of puromycin. The cells that still emit green fluorescence and survive 8 days later are considered successful lentivirus packaging. After three generations, the single-cell lines were isolated and continued to culture and passage to obtain a monoclonal cell line containing the N protein, which was named CRFK-PHBLV-N. Western blot, IFA, and other experiments were executed to validate its construction. The FIPV-CQ strain was used to infect both the stable cell lines and CRFK cells. The replication capacity of the virus in the two cell types was compared through CPE, RT-qPCR, TCID50, Western blot, and IFA detection.

## Cell viability assay

The reserve solution of UDCA was diluted into six concentrations of 20, 40, 60, 100, 200, and 400 µg/mol with EMEM basal medium. When the cells grew to 85%–90% confluence in the culture wells, the UDCA was added. One hundred microliters of completed diluted UDCA solution was added to each well, and the cells were incubated at 37°C for 24 h in the incubator. At the end of incubation, 10 µL of CCK-8 solution was added to each well, and cells were incubated for 1 h. After incubation, the absorbance value was measured at 450 nm using an enzyme label analyzer to assess the activity state of the cell. The percentage of cell viability (%) was calculated as follows: cell viability = $[(As - Ab)/(Ac - Ab)] \times 100\%$, where "As" is the absorbance of experimental wells (with cells, medium, CCK-8, and different concentrations of drugs), "Ac" is the absorbance of control wells (with cells, medium, and CCK-8), and "Ab" is the absorbance of blank wells without cells (with medium and CCK-8).

## Western blot

After treating the cells with appropriate drugs or viruses, 200 µL of pre-cooled RIPA lysis buffer containing a protease inhibitor mixture was added to each well to extract total cellular proteins, and RIPA lysis buffer containing the mixture of phosphatase inhibitor was used to extract cellular phosphorylated proteins. Protein samples were subjected to 8% SDS-PAGE electrophoresis, and after cutting the gel according to the Marker, the target proteins were electroblotted and transferred onto a polyvinylidene difluoride membrane. The membranes were blocked with TBST solution containing fast protein blocking solution, washed with TBST, and then incubated with the corresponding antibodies overnight at 4°C. After TBST washing, the membranes were incubated with the appropriate horseradish peroxidase-coupled secondary antibody for 1 h at room temperature. After TBST wash, the protein bands were detected using ECL detection reagent. The relative expression level of the target protein was quantified using ImageJ software, and β-tubulin or GAPDH was used as an internal reference protein for normalization.

## Indirect immunofluorescence assay

CRFK cells and CRFK-PHBLV-N cells were seeded in 24-well plates containing cell crawling. After different treatments, the cells were fixed with 4% paraformaldehyde for 30 min and then permeabilized with 0.1% Triton X-100 at room temperature for 10 min. Then, the cells were blocked with 5% BSA at room temperature for 1 h. After washing, primary antibodies (anti-FIPV-N antibody, anti-IRF3 antibody, and anti-STAT1 antibody) were added and incubated at 4°C overnight. Subsequently, the cells were washed with PBS and incubated with a goat anti-rabbit fluorescent secondary antibody for 1 h at room temperature in the dark. Finally, DAPI was added for 10 min to stain the nuclei. Cell crawler samples were analyzed by indirect immunofluorescence imaging using an inverted laser confocal microscope (model: LSM 800, ZEISS).

## Virus inactivation, attachment, entry, replication, and release assay

Direct inactivation group: FIPV (MOI = 1) was mixed with UDCA$_{200\mu g/mol}$ and incubated at 37°C for 3 h. The cells were pre-cooled at 4°C for 1 h, then the incubated mixture was added to the pre-cooled cells in a homogeneous dropwise manner, and the cells were incubated at 4°C for 2 h. When the incubation was completed, the cells were gently washed three times with PBS, and the wells were supplemented with 2 mL EMEM basal medium and continued to be cultured at 37°C, 5% $CO_2$ incubator for 36 h.

Adsorption group: CRFK-PHBLV-N cells were pre-treated with 200 µg/mol UDCA at 37°C for 1 h. Subsequently, a mixture of UDCA $_{200\mu g/mol}$ and FIPV (MOI = 1) was added, and the cells were incubated at 4°C for 1 h.

Internalization group: CRFK-PHBLV-N cells were pre-cooled at 4°C for 1 h, and then FIPV (MOI = 1) was added and incubated at 4°C for 1 h. Then the cells were thoroughly washed with pre-cooled PBS to remove unbound viral particles, cultured with EMEM medium containing UDCA at a concentration of 200 µg/mol, and incubated at 37°C for 2 h.

Replicative group: CRFK-PHBLV-N cells were infected with FIPV (MOI = 1) and incubated at 37°C for 6 h. The viral solution was discarded and washed three times with PBS, followed by the addition of UDCA$_{200\mu g/mol}$ for 10 h.

Release group: After infecting CRFK-PHBLV-N cells with FIPV (MOI = 1) for 16 h, the EMEM culture medium with a concentration of UDCA$_{200\mu g/mol}$ was added and incubated at 37°C for 1 h. The supernatant from the above steps was collected and used to detect FIPV by fluorescence PCR. Quantitative PCR was used to detect the copy number of FIPV N gene, and all cell samples were collected to detect the changes in the expression of FIPV N protein.

## Transmission electron microscopy

The collected viral fluid was concentrated according to the instructions of the Viral Fluid Concentration Kit. Negative staining was performed on the UDCA-treated viral liquid samples by taking a drop of about 20 µg/mol of the sample onto a copper mesh, and the excess liquid was absorbed with filter paper after 3 min, then placed on 2% phospho-tungstic acid staining solution for 1–2 min, and absorbed again. A JEM-1400FLASH transmission electron microscope of Japan Electronics Corporation (JEOL) was used to image the acquisition of the copper mesh to observe the morphological changes of the virus particles.

## Real-time fluorescent quantitative PCR assay

Quantitative reverse transcription PCR (RT-qPCR) was used to detect FIPV-N RNA transcript levels in CRFK-PHBLV-N cells. Total RNA was extracted using TRIzol reagent according to the manufacturer's instructions, and cDNA was synthesized using the PrimeScript RT Kit (TaKaRa, Japan). cDNA was subsequently analyzed by RT-qPCR using 2× Universal SYBR Green Fast qPCR Mix and specific primers (Table 1). Absolute qPCR

TABLE 1 Information on RT-qPCR primers for FIPV-N[a]

| Primer names | Sequences | Fragment size |
|---|---|---|
| FIPV-N-F | 5′-ACAGGGACAACGCGTCAACT-3′ | 286 bp |
| FIPV-N-R | 5′-TCCTGTACCCAAGAAGTAGAAGAAC-3′ | |

[a]F, forward; R, reverse.

assays were performed using specific primers FIPV N-forward and FIPV N-reverse to measure FIPV N copy number.

## Transcriptomic screening of UDCA anti-FIPV related signaling pathway

In order to investigate the innate immune responses occurring in hosts after UDCA treatment and FIPV infection, transcriptomic analysis was executed to screen the relevant signaling pathways in CRFK-PHBLV-N cells co-treated with 200 µg/mol of FIPV-UDCA, and further verified the regulatory mechanism of these pathways, which provided a theoretical basis and reference for the response strategy of host innate immunity when infected with FIPV.

## Statistical analysis

All statistical analyses were performed using GraphPad Prism 8.0 (GraphPad Inc., La Jolla, CA, USA). All data are presented as the means ± standard deviations or mean the standard error of the mean of three independent experiments. One-way analysis of variance and $t$-tests were used to determine the statistical differences among multiple groups. $P$ values <0.05 were considered statistically significant (significance in the figures is indicated as follows: *, $P < 0.05$; **, $P < 0.01$; ***, $P < 0.001$; and ****, $P < 0.0001$).

## RESULTS

### Successful construction of stable cell lines

The recombinant lentiviral plasmids were transfected to HEK293T cells, and both the PHBLV and PHBLV-N groups showed the ZsGreen fluorescence (Fig. 1A). The lentivirus was packaged (Fig. 1B) and verified by Western blot assays (Fig. 1C). The optimal concentration of puromycin on CRFK cells was determined, and the result showed that when the concentration of puromycin was 1 µg/mL, all the cells died on the eighth day (Fig. 1D). After the packaged lentivirus infected the CRFK cells, significant fluorescence was observed at 96 h (Fig. 1E), then the pressure screening was performed with puromycin and detected the FIPV N of the CRFK cells by Western blot (Fig. 1F). The results showed that the N protein can be detected, demonstrating that the lentiviral packaging was successful. Then, the stability of N protein expression in different generations (P1 to P30) of CRFK-PHBLV-N cells (Fig. 1G) was assessed. The IFA detected the N protein expression (Fig. 1H), and the result was consistent with that of the Western blot. The monoclonal cell line was named CRFK-PHBLV-N.

### Stabilization of cell lines enhances FIPV infection titers

According to the CPE observation in CRFK cells and CRFK-PHBLV-N cells infected with FIPV, compared to the CRFK cells group, the cytopathy in the CRFK-PHBLV-N cells group was more obvious (Fig. 2A and B). Transmission electron microscopy observations found that the stable cell line group contained more virus particles after infecting with FIPV (Fig. 2C). The results of TCID50 (Fig. 2D), RT-qPCR (Fig. 2E), Western blot (Fig. 2F and G), and IFA (Fig. 2H) showed that a significantly higher viral titer, viral copy number, and protein expression in CRFK-PHBLV-N cells than those of CRFK cells after infecting with FIPV. The viral titer increased nearly 10 times, suggesting that the CRFK-PHBLV-N cells could effectively promote the replication of FIPV.

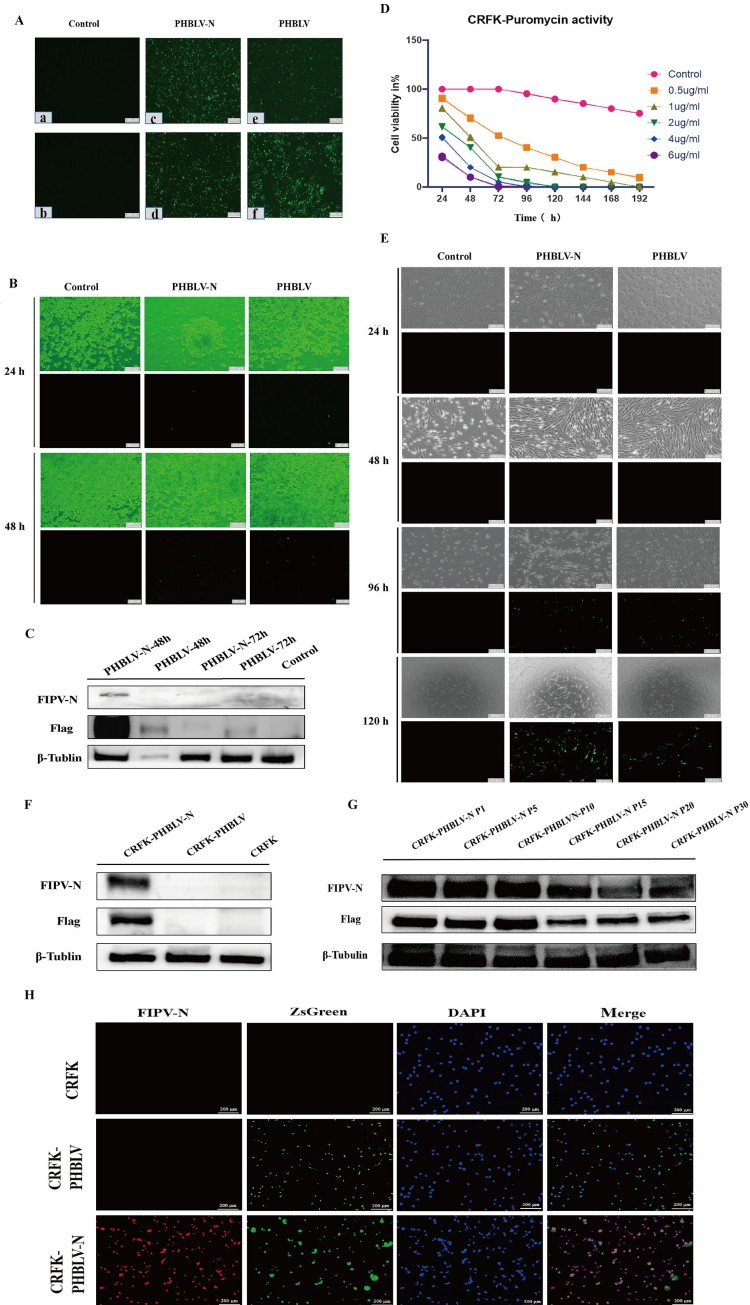

FIG 1 Construction and validation of stable cell lines. (A) The lentivirus recombinant plasmid transfection of HEK293T cells. (B and C) Observation of lentivirus fluorescence signals and identification of protein expression levels. (D) The optimal death curve of cells caused by puromycin. (E) Pressure screening of stable cell lines. (F) Detection of N protein expression in monoclonal cells by Western blot. (G) Stability detection of the monoclonal cells by Western blot. (H) The detection of N protein expression in the monoclonal cell by IFA identification.

## UDCA co-treatment shows a significant inhibitory effect on FIPV infection *in vitro*

In order to evaluate the inhibitory effect of UDCA on FIPV *in vitro* and to determine its optimal working concentration and treatment, we chose concentration gradients of 10 µg/mol, 20 µg/mol, 40 µg/mol, 60 µg/mol, 100 µg/mol, 200 µg/mol, and 400 µg/mol for CCK-8 experiment of CRFK-PHBLV-N cells, the results showed that the survival rate of

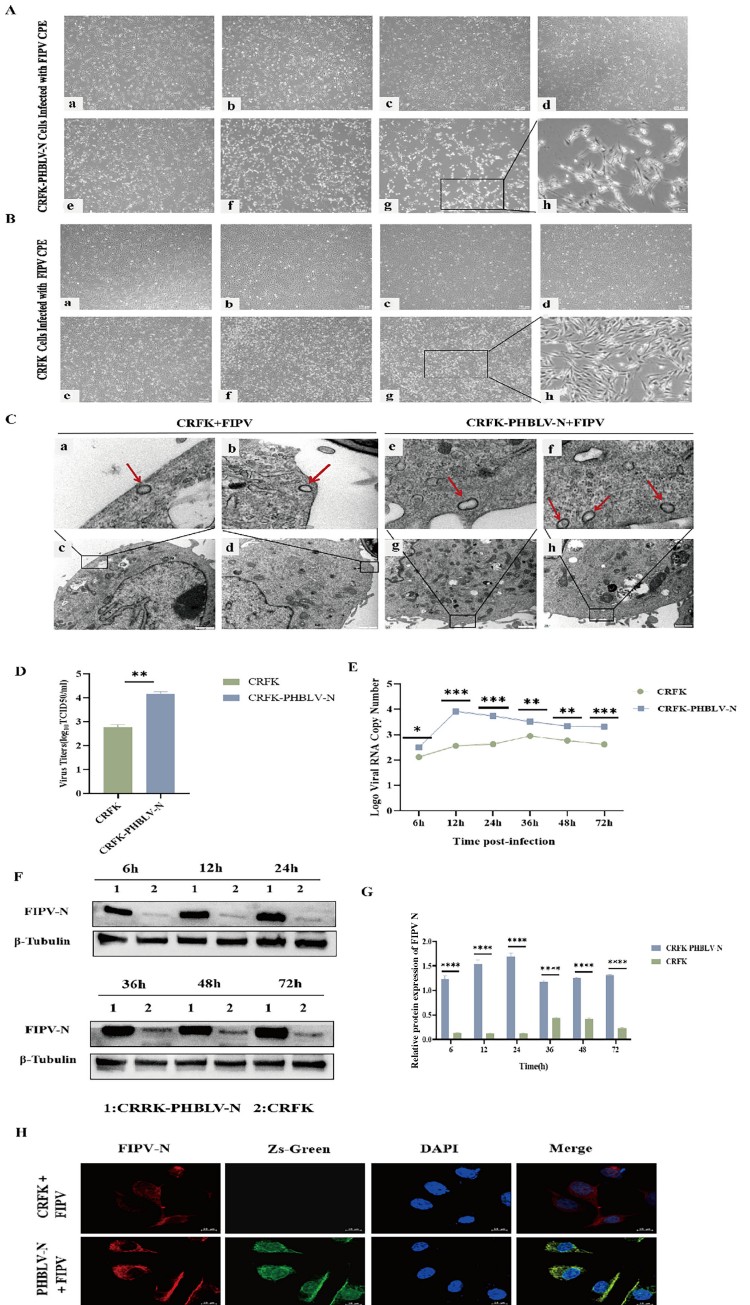

**FIG 2** Proliferative ability of CRFK and CRFK-PHBLV-N cells to FIPV. (A) CPE of FIPV-infected CRFK-PHBLV-N cells at different time points: a, negative control, normal cell morphology; b, 6 h, no obvious change in cell morphology; c, 12 h, cells were stretched; d, 24 h, the number of dead cells increased; e, 36 h, part of the cells appeared to die and fall off; f, 48 h, cells showed massive death, vacuoles, and crumpling; g, 72 h, nuclear consolidation, no cell morphology; h, local magnification of g, cells were stretched, crumpled, and no cell morphology. (B) CPE of FIPV-infected CRFK cells at different time points: a, negative control, normal cell morphology; b, 6 h, no significant change in cell morphology; c, 12 h, no significant change in cell morphology; d, 24 h, increased number of dead cells; e, 36 h, some cells appeared to be dead; f, 48 h, some cells appeared to be dead; g, 72 h, a large portion of cells appeared to be dead but with fewer detached; h, local amplification of g, cells stretched and crumpled, and no cell morphology. Localized enlargement of FIPV, cells are stretched and have cellular morphology. (C) FIPV-positive staining transmission electron micrographs. a–d, CRFK cells infected with FIPV group; e and f, CRFK-PHBLV-N cells infected with FIPV group; a, local magnification of b; c, local magnification of d; e, local magnification of

**Fig 2 (Continued)**

g; f, local magnification of h. (D) Quantification of FIPV-CQ TCID50. (E) RT-qPCR detection of viral copy number and growth curves in both cells. (F and G) Expression level and quantitative analysis of FIPV-N protein in different cells. (H) IFA detection of FIPV infection with CRFK and CRFK-PHBLV-N. *$P < 0.05$; **$P < 0.01$; ***$P < 0.001$; and ****$P < 0.0001$.

CRFK-PHBLV-N cells treated with UDCA at the concentration of 200 µg/mol was closest to 100% (Fig. 3A). Therefore, in order to maximize the recovery of the normal physiological environment of the cells and avoid the interference of drug toxicity on the test results, 200 µg/mol was finally selected as the optimal treatment concentration of UDCA for the subsequent study. This concentration was chosen to ensure that the cells maintained normal physiological state during drug treatment, so as to more accurately evaluate the inhibitory effect of UDCA on FIPV infection and its potential mechanism. The drug was administered at this concentration in three delivery modes: pre-treatment, co-treatment, and post-treatment. The results of RT-qPCR (Fig. 3B), Western blot (Fig. 3C and D), and IFA (Fig. 3E and F) showed that co-treatment with UDCA at a concentration of 200 µg/mol was the optimal and most stable treatment method.

## UDCA inhibits viruses at the stages of viral inactivation, replication, and internalization and disrupts the viral particle structure of FIPV

To explore the effect of UDCA on FIP infection at different stages, a direct inactivation group, an adsorption treatment group, an internalization treatment group, a replication treatment group, and a release treatment group were set up, and samples from each group were collected separately (Fig. 4A). The copy number of the FIPV-N gene in the supernatant of cells and in CRFK-PHBLV-N cells was detected by absolute RT-qPCR. It was found that UDCA treatment in the direct inactivation group and the replication treatment group significantly reduced the copy number of the FIPV-N gene in the CRFK-PHBLV-N cells (Fig. 4B). The results of the Western blot also showed that UDCA could significantly reduce the expression level of FIPV-N protein in both the direct inactivation period and the replication period (Fig. 4C and D). To further verify whether UDCA could directly destroy the structure of FIPV virus particles, we performed phosphotungstic acid negative staining on the concentrated viral solution and observed it under a transmission electron microscope. The results (Fig. 4E) showed that the virions in the FIPV group were morphologically intact, round, and radially arranged fibrils with a diameter of about 100 nm, which was in line with the typical characteristics of coronavirus particles. The structural integrity of the virus particles in the FIPV + UDCA-treated group was damaged, and the internal substance of the viral particles partially flowed out. This suggests that UDCA has the ability to directly destroy the structure of FIPV viral particles, thereby affecting their effectiveness in infecting host cells. In summary, the results of this study indicated that UDCA not only directly destroys the structure of FIPV viral particles but also significantly inhibits their replication during the viral replication phase of the virus.

## Transcriptomic data show that UDCA processing is closely related to the cellular innate immune signaling pathway

Based on the Kyoto Encyclopedia of Genes and Genomes (KEGG) database, the differentially expressed genes (DEGs) of the FIPV-infected group and the FIPV-UDCA$_{200µg/mol}$ co-treated group were systematically analyzed for pathway enrichment analysis using ClusterProfiler software (Fig. 5A). After Bonferroni correction ($P < 0.05$), the results showed that DEGs were significantly enriched in several key pathways to immune defense: (i) the IL-17 signaling pathway and the JAK-STAT signaling pathway, as the core of cytokine signaling pathways, regulate the activation of innate immune and inflammatory response; (ii) natural killer cell-mediated cytotoxicity, as an important effect mechanism of intrinsic immunity, directly participates in the clearance of virus-infected

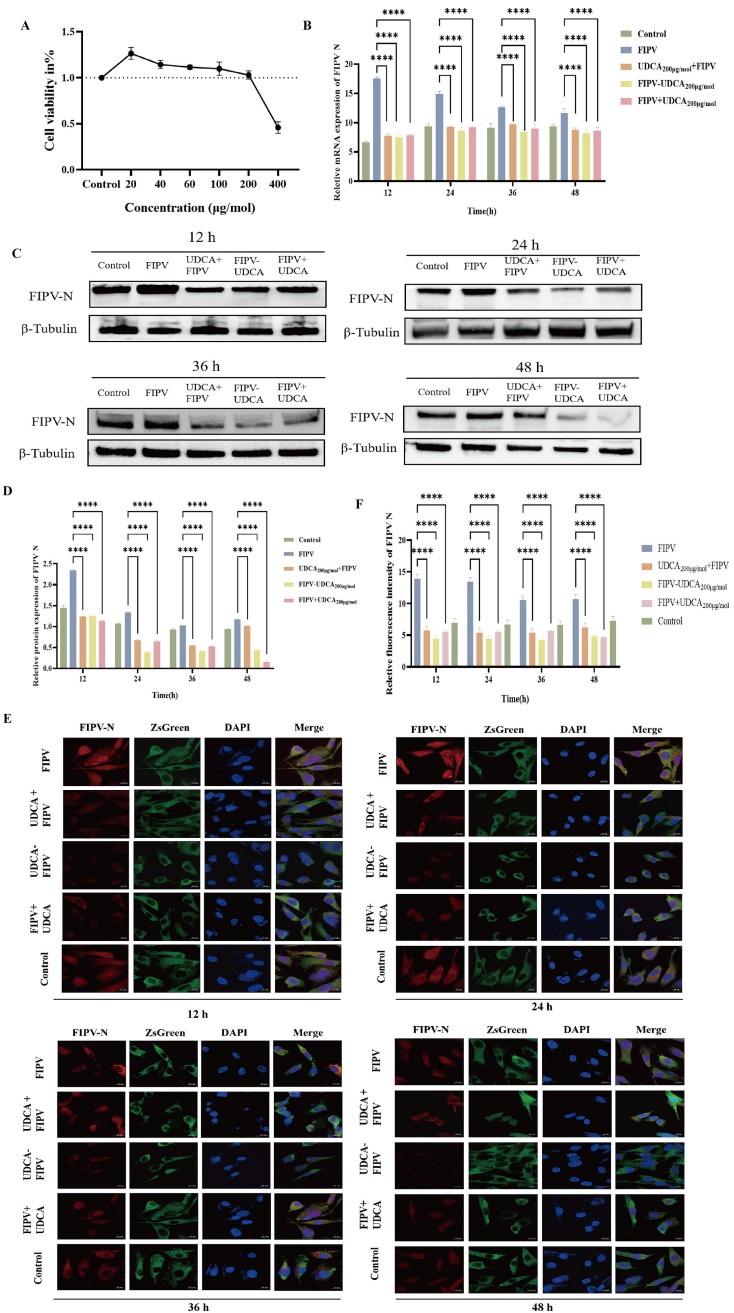

**FIG 3** Validation of the optimal working concentration and treatment of UDCA to inhibit FIPV proliferation. (A) CCK-8 assay for the effect of ursodeoxycholic acid on the activity of CRFK-PHBLV-N cells. (B) FIPV N gene level after the action of ursodeoxycholic acid. (C) Western blotting to detect the changes of FIPV N protein. (D) Gray scale of the FIPV N protein change analysis. (E) CRFK-PHBLV-N cells in FIPV IFA results graph. (F) CRFK-PHBLV-N cells in FIPV fluorescence intensity analysis. ****$P < 0.0001$.

cells; and (iii) the vascular endothelial growth factor (VEGF) signaling pathway plays an important role in regulating the immune microenvironment. Since the previous studies have shown that UDCA is closely related to innate immunity, the subsequent studies mainly focused on the innate immunity pathway. According to the volcano plotting results (Fig. 5B), 130 genes were upregulated, and 425 genes were downregulated. Further visualization of the Circos revealed that UDCA treatment specifically activated the core components of the JAK-STAT pathway and its downstream effector molecules, MX1 and ISG15, which were upregulated. Correlation analysis of antiviral

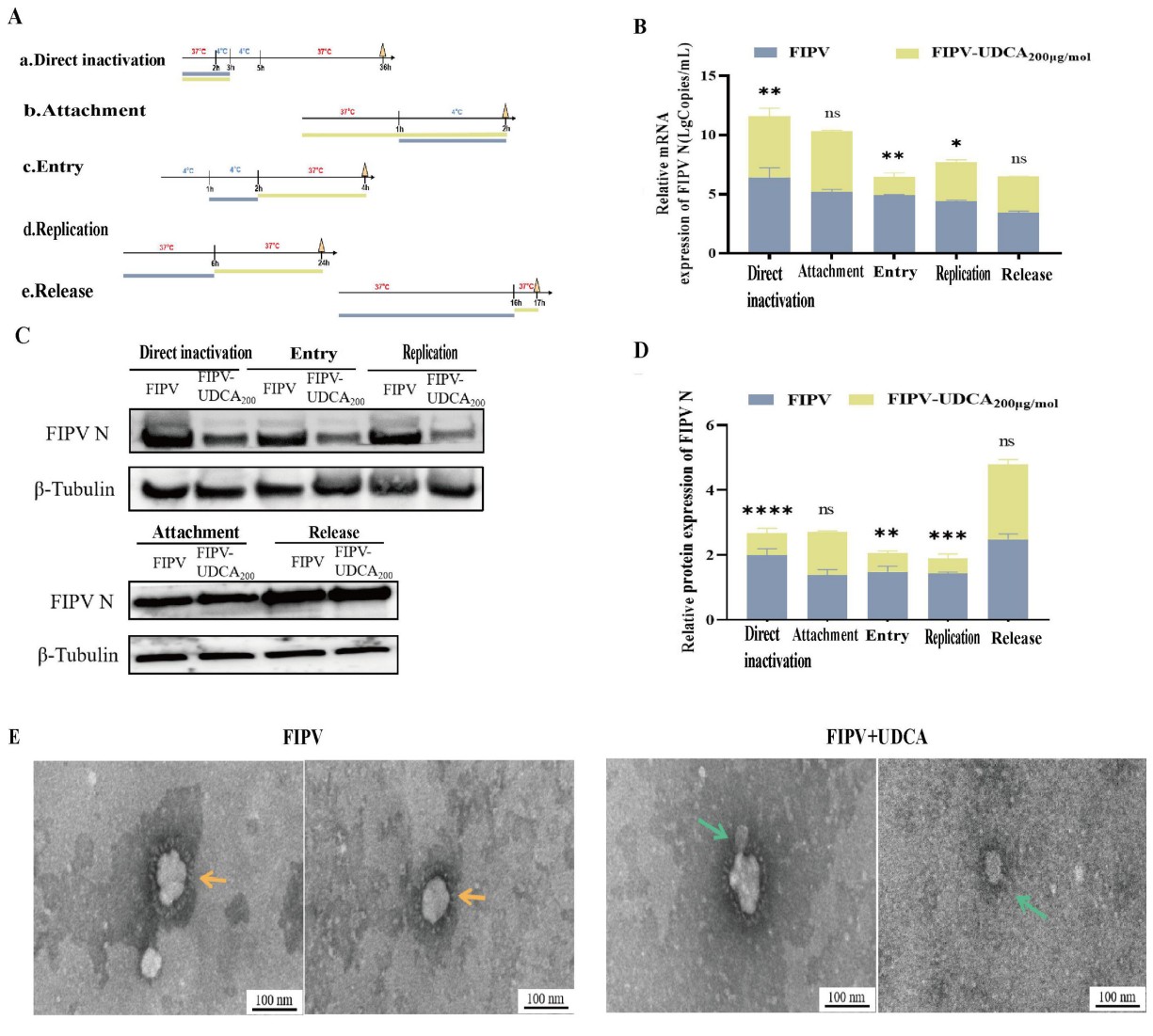

**FIG 4** Changes in FIPV N gene copy number and protein expression at different stages after ursodeoxycholic acid treatment. (A) Graph of different stages of treatment. (B) Graph of FIPV N gene copy number results in CRFK-PHBLV-N cells at different stages. (C) Graph of FIPV results in CRFK-PHBLV-N cells at different stages. (D) Grayscale analysis of FIPV in CRFK-PHBLV-N cells at different stages. (E) Negative staining electron micrograph of FIPV virus particles. *$P < 0.05$; **$P < 0.01$; ***$P < 0.001$; ****$P < 0.0001$; and ns, not significant.

gene expression (Fig. 5C) confirmed that, compared with the FIPV-only infected group, the STAT1, TLR3, IFNB1, IRF9, and interferon-stimulated genes (ISGs) were significantly upregulated in the UDCA$_{200\mu g/mol}$ co-treatment group ($P < 0.05$), which formed a synergistically activated cascade amplification network in the JAK-STAT pathway, significantly enhancing the cellular antiviral status.

IFN-β of type I interferon family, due to its high functional similarity with IFN-α and its lack of antagonism by certain viral proteins such as FCoV ORF7a, can more clearly reflect the host's natural immune response. The role of IFN-β in antiviral immunity has been widely recognized, especially in therapeutic strategies targeting the immune escape mechanism of viruses, where IFN-β may become a key therapeutic target. Moreover, the significant upregulation factor of IFNB1 was present, because the changes in IFN-β secretion level in CRFK-PHBLV-N cells after UDCA treatment showed that the IFN-β secretion level was lower in the FIPV-infected group compared with the control group at 24 h and 36 h. The IFN-β secretion level was higher in the UDCA-treated group

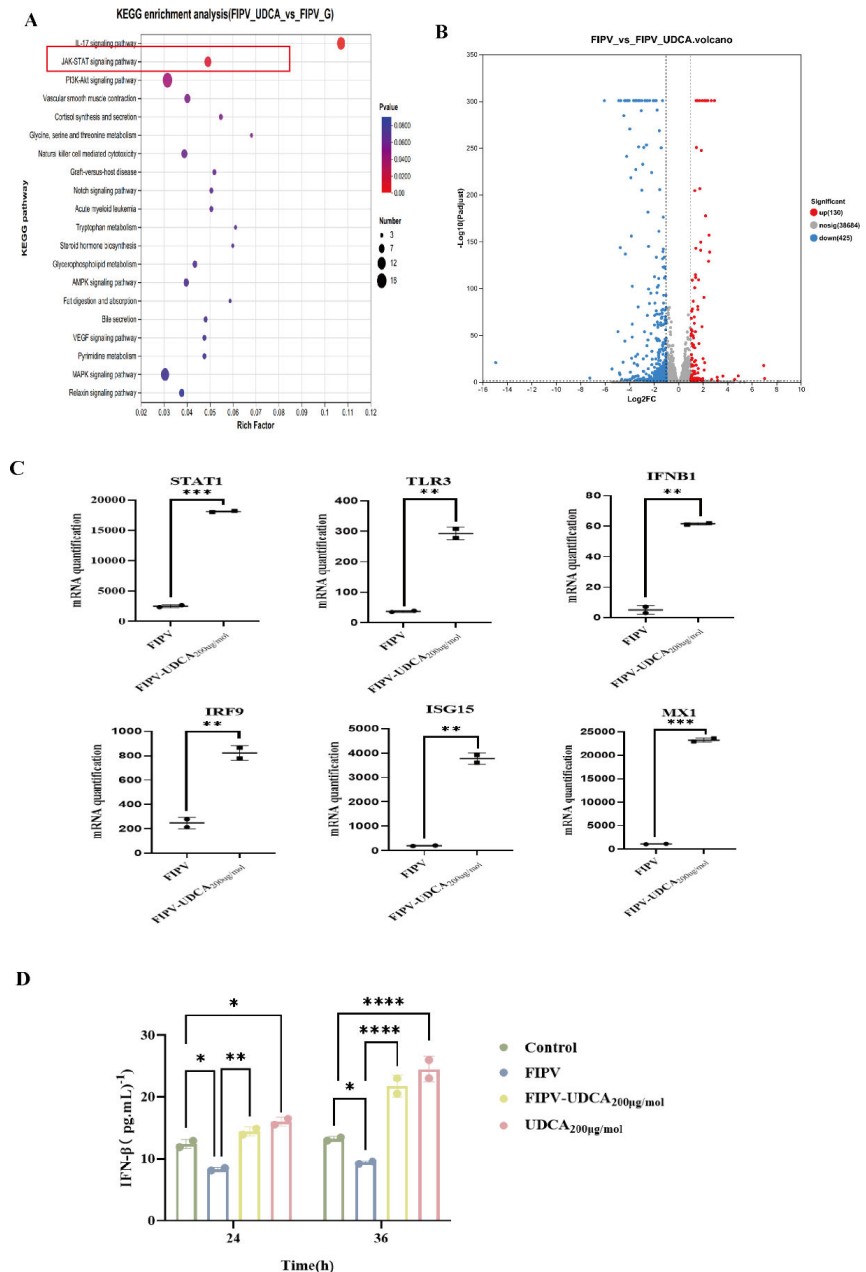

**FIG 5** Screening of signaling pathways associated with anti-FIPV infection after UDCA co-treatment based on transcriptomic data. (A) KEGG pathway enrichment analysis. (B) Differential gene volcano plot. (C) Correlation analysis of FIPV-UDCA$_{200\ \mu g/mol}$ co-treatment with antiviral gene expression. (D) IFN-β ELISA results in CRFK-PHBLV-N cells. *$P$ < 0.05; **$P$ < 0.01; ***$P$ < 0.001; and ****$P$ < 0.0001.

and the FIPV-UDCA$_{200\ \mu g/mol}$ co-treated group at 24 h and 36 h, and the secretion level was higher in the UDCA-treated and FIPV-UDCA$_{200\ \mu g/mol}$ co-treated groups. The secretion level of IFN-β in the UDCA-treated group and the FIPV-UDCA$_{200\mu g/mol}$ co-treated group was significantly higher at 24 h and 36 h (Fig. 5D) ($P$ < 0.01). The results indicated that FIPV infection could inhibit IFN-β secretion to a certain extent, thus promoting viral replication, whereas UDCA treatment stimulated the secretion of IFN-β from CRFK-PHBLV-N cells, thus antagonizing the inhibitory effect of FIPV on the secretion of IFN-β by the cells, and thus enhancing the immune defense effect.

## UDCA significantly activates STAT1 phosphorylation and promotes its nuclear translocation

Under physiological conditions, STAT1 protein exists in an inactive form in the cytoplasm. When FIPV invades host cells, it effectively inhibits the activation of the JAK-STAT signaling pathway by specifically inhibiting the tyrosine phosphorylation of STAT1 (especially the Tyr701 site) and blocking its nuclear translocation process. This immune escape mechanism creates a favorable intracellular environment for viral replication. At the molecular level, FIPV can directly interact with STAT1 or regulate the expression of interferon receptor, disrupting the JAK-STAT signaling cascade, through non-structural proteins such as nsp1. To further verify whether the JAK-STAT signaling pathway was activated after UDCA treatment, we set up a blank control group, an FIPV-infected group, an FIPV-UDCA $_{200 \ \mu g/mol}$ co-treated group, and a UDCA-alone group, and collected cell protein samples from the cells for 24 h and 36 h, and detected the phosphorylation level of STAT1 in the cells by Western blot, and analyzed the phosphorylation level of STAT1 in the cells after UDCA treatment by IFA. Western blot was used to detect the STAT1 phosphorylation in cells, and IFA was used to analyze the changes in the STAT1 nuclear translocation after UDCA treatment.

The results showed that compared with the control group, the levels of STAT1 phosphorylation and nuclear translocation were significantly reduced at 24 h and 36 h after FIPV infection in the UDCA-treated group, while STAT1 protein phosphorylation levels (Fig. 6A and B) and nuclear translocation levels (Fig. 6C and D) were significantly elevated at 24 h and 36 h in the UDCA-treated group and the FIPV-UDCA$_{200 \ \mu g/mol}$-treated group ($P < 0.0001$); this suggested that viral infection inhibited the level of phosphorylated STAT1 in cells, whereas UDCA enhances the level of phosphorylated STAT1 and activates JAK ($P < 0.0001$), which indicated that virus infection inhibited the intracellular level of phosphorylated STAT1, while UDCA could enhance the level of STAT1 phosphorylation and nuclear translocation, and then activate the JAK-STAT signaling pathway to play a role in immune defense.

## UDCA enhances the function of upstream regulators by upregulating TLR3 expression and promoting IRF3 activation

To further elucidate the role of UDCA on the upstream regulation of JAK-STAT signaling pathway, this study analyzed the effects of UDCA on the expression of TLR3 protein and the activation status of IRF3 by a systematic assay at the blank control group, FIPV-infected group, FIPV-UDCA $_{200 \ \mu g/mol}$ co-treatment group, and UDCA alone treatment group were set up in the experiment, and the samples were collected at 24 h and 36 h. Western blot was used to detect the expression level of TLR3 protein and the degree of phosphorylation of IRF3, and IFA was used to observe the nuclear translocation of IRF3.

The results showed that FIPV infection significantly inhibited TLR3 protein expression (Fig. 7A and B), IRF3 phosphorylation level (Fig. 7A and C), and nuclear translocation (Fig. 7D and E) compared with those of the blank control group ($P < 0.001$). Notably, UDCA treatment effectively reversed these FIPV-induced inhibitory effects. In both the FIPV-UDCA $_{200 \ \mu g/mol}$ co-treatment group and the UDCA alone treatment group, TLR3 protein expression, IRF3 phosphorylation, and nuclear translocation efficiency showed a significant rebound ($P < 0.001$). These results confirmed that UDCA not only regulates the core components of the JAK-STAT signaling pathway, but also enhances the function of upstream regulators by upregulating TLR3 expression and promoting IRF3 activation.

## UDCA improves host's antiviral defense system by enhancing the expression of downstream effector molecules ISG15 and MX1

In order to systematically evaluate the regulatory effects of UDCA on the downstream effector molecules of JAK-STAT signaling pathway, the present study focused on the expression changes of ISG15 and MX1 in ISGs. The experimental design included four treatment groups: blank control group, FIPV-infected group, FIPV-UDCA $_{200 \ \mu g/mol}$

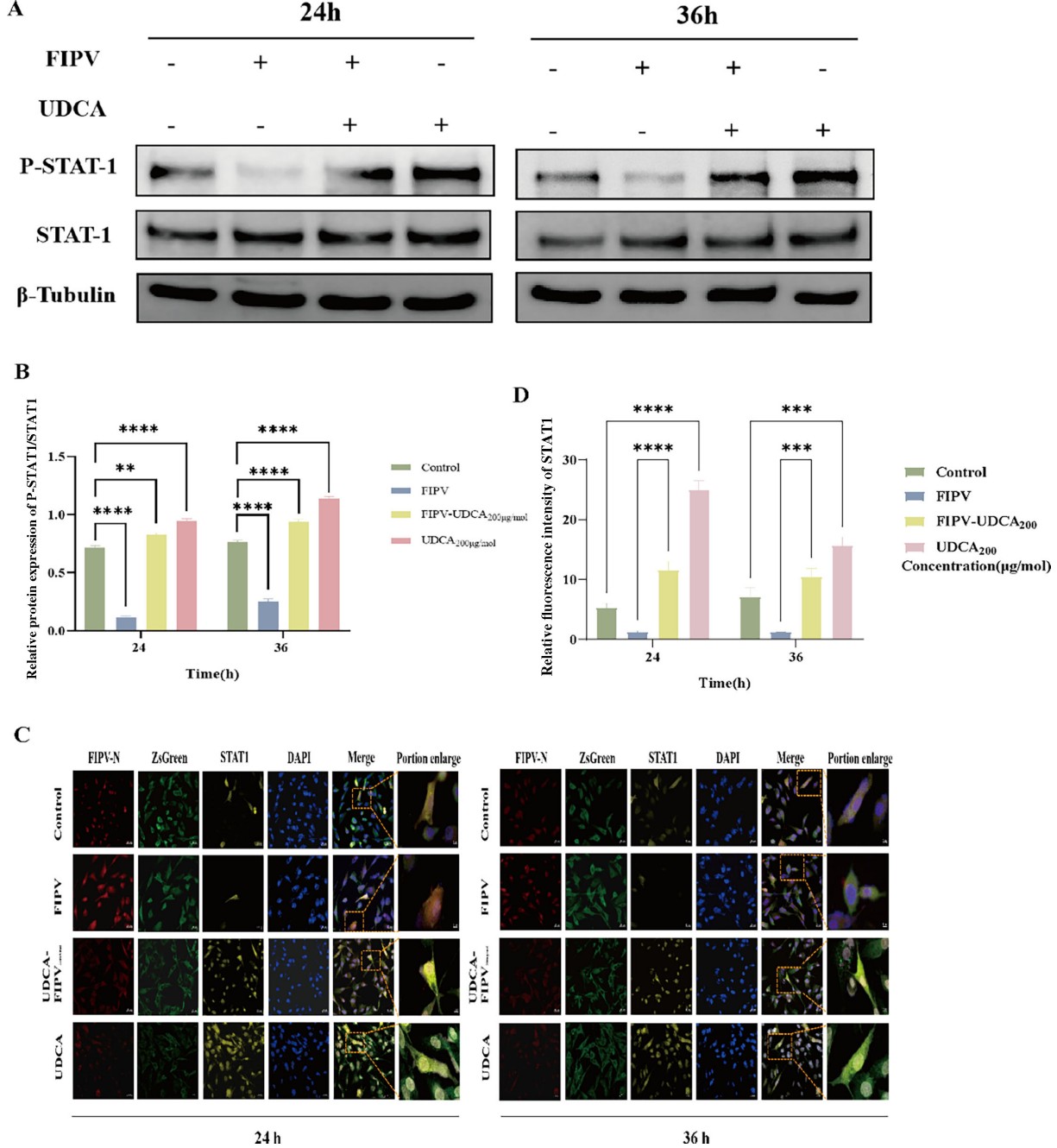

**FIG 6** Changes in STAT1 phosphorylation and nuclear translocation levels in CRFK-PHBLV-N cells after ursodeoxycholic acid treatment. (A) STAT1 Western blot results in CRFK-PHBLV-N cells. (B) Grayscale analysis of p-STAT1/STAT1 protein in CRFK-PHBLV-N cells. (C) CRFK-STAT1 IFA results in PHBLV-N cells. (D) STAT1 fluorescence intensity analysis in CRFK-PHBLV-N cells. **$P < 0.01$; ***$P < 0.001$; and ****$P < 0.0001$.

co-treatment group, and UDCA alone treatment group, and the cell samples were collected at 24 h and 36 h for Western blot analysis.

The results showed that FIPV infection significantly inhibited the protein expression of ISG15 and MX1 (Fig. 8A through C), and the expression levels were significantly downregulated compared with the control group at both 24 h and 36 h time points ($P < 0.001$). However, after UDCA treatment, both the co-treated group and the group treated with FIPV alone significantly reversed the inhibitory effect, and the expression levels of ISG15 and MX1 were effectively restored at 24 h and 36 h ($P < 0.05$). These data suggested that UDCA not only activated the upstream regulators of JAK-STAT signaling

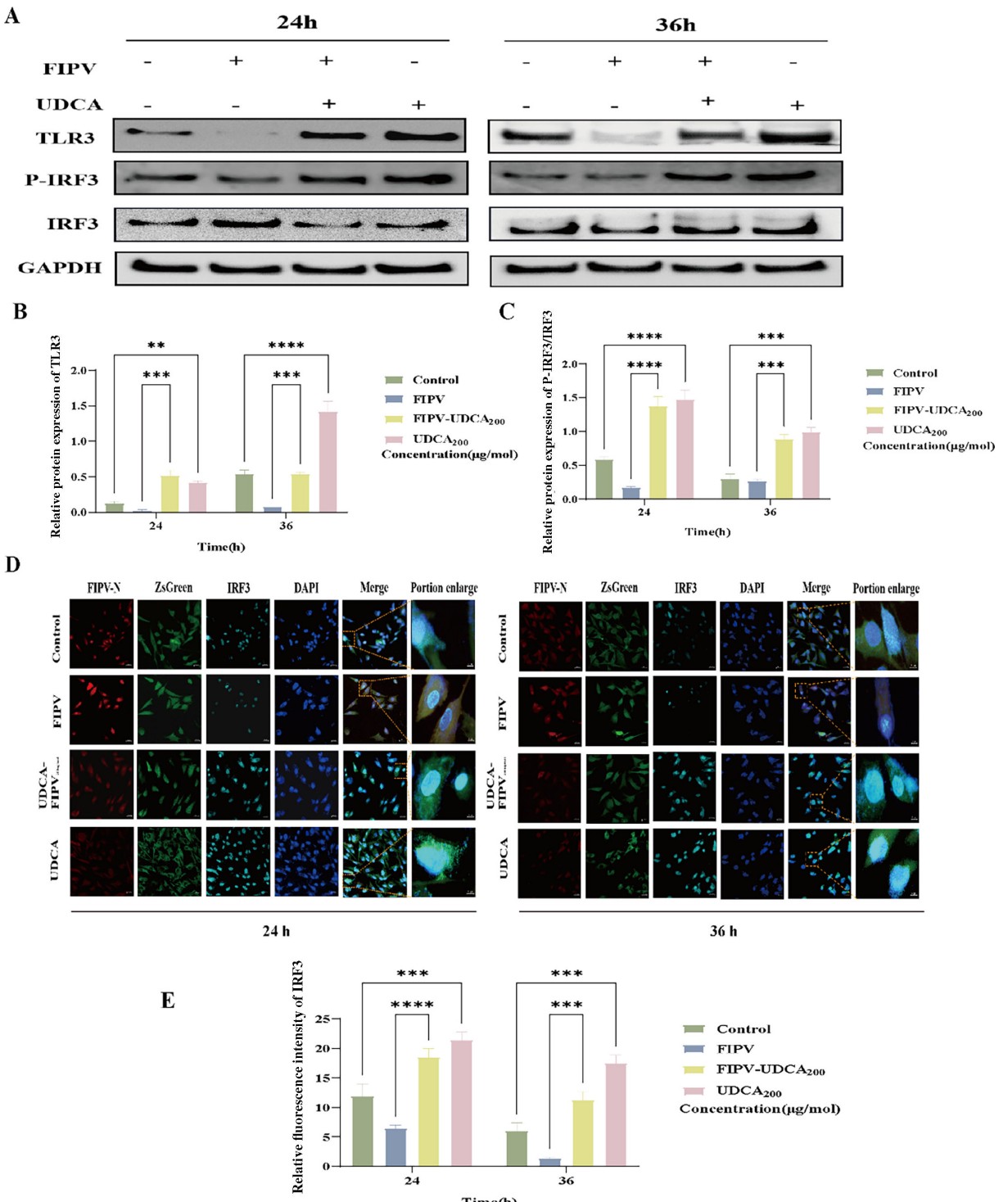

**FIG 7** Changes in TLR3 protein, IRF3 phosphorylation, and nuclear translocation levels in CRFK-PHBLV-N cells after ursodeoxycholic acid treatment. (A) Graph of TLR3 and IRF3 Western blot results in CRFK-PHBLV-N cells. (B) Graph of TLR3 protein grayscale analysis in CRFK-PHBLV-N cells. (C) Grayscale analysis plot of p-IRF3/IRF3 protein in CRFK-PHBLV-N cells. (D) Graph of IRF3 IFA results in CRFK-PHBLV-N cells. (E) Graph of IRF3 fluorescence intensity analysis in CRFK-PHBLV-N cells. **$P < 0.01$; ***$P < 0.001$; and ****$P < 0.0001$.

pathway, but also enhanced the expression of downstream effector molecules, ISG15 and MX1, and further improved the host's antiviral defense system.

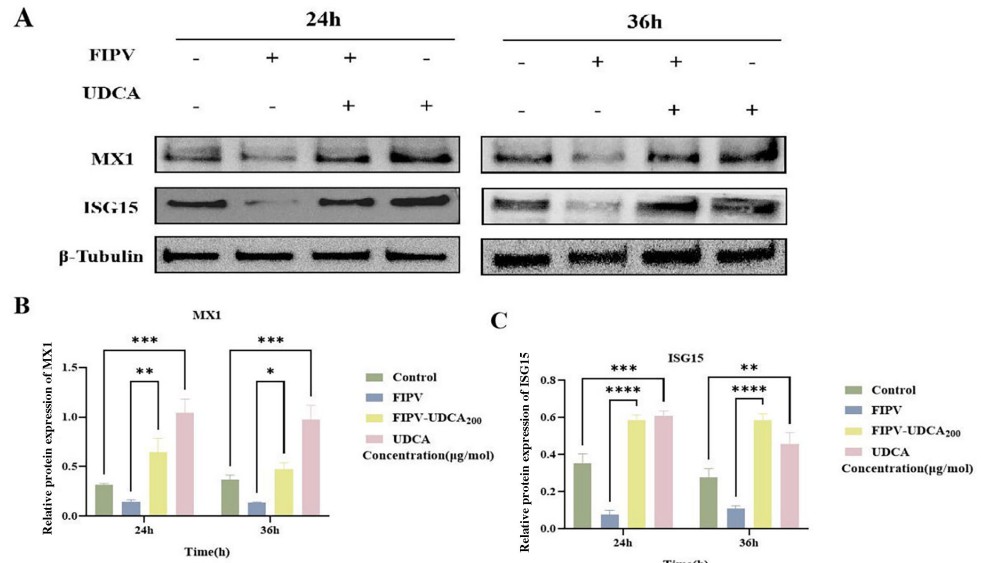

**FIG 8** Changes in MX1 and ISG15 protein levels in CRFK-PHBLV-N cells after ursodeoxycholic acid treatment. (A) Western blot results of MX1 and ISG15 in CRFK-PHBLV-N cells. (B) Grayscale analysis of MX1 protein in CRFK-PHBLV-N cells. (C) Grayscale analysis of ISG15 protein in CRFK-PHBLV-N cells. ISG15 protein grayscale analysis graph in CRFK-PHBLV-N cells. $*P < 0.05$; $**P < 0.01$; $***P < 0.001$; and $****P < 0.0001$.

## DISCUSSION

FIP, as a highly lethal infectious disease caused by FCoV, has become one of the key diseases in the field of animal medicine. With the development of the social economy and changes in people's lifestyles, the number of pet cats kept has shown a rapid growth trend in recent years, especially as the popularity of purebred cats continues to rise. However, influenced by a variety of environmental factors, the prevalence of FCoV in cat populations has increased significantly, and the number of clinical cases has continued to increase. With the continuous emergence and prevalence of FIPV variants, the development of vaccines faces many challenges, and existing inhibitors such as GS441 are expensive and have a long treatment period, making the development of highly effective and economical anti-FIPV small molecule drugs particularly urgent. In recent years, it has been found that baicalin may inhibit FIPV infection by modulating the PI3K-AKT pathway and apoptotic pathway (11). At safe doses, natural compounds such as chymotrypsin, quercetin, and stigmasterol may have inactivating effects on FIPV. A study by Tomoyoshi Doki's team at Kitasato University showed that GS-441524 synergistically inhibited FIPV in combination with itraconazole (Science, 2022) (12). Based on the need for multi-targeted interventions for FIPV treatment, the potential value of UDCA as a compound with broad-spectrum antiviral activity is worth exploring. UDCA is not only able to inhibit a variety of viruses, such as hepatitis B virus, hepatitis C virus, and human immunodeficiency virus (HIV) by regulating host-associated proteins and signaling pathways, but it also exerts a significant inhibitory effect on the same coronavirus family of SARS-CoV-2 and transmissible gastroenteritis virus. Based on this, it is of great scientific significance and application value to investigate whether UDCA has an anti-FIPV effect and its mechanism of action.

The mechanism of UDCA action against coronaviruses (such as SARS-CoV-2 and FIPV) is mainly explored from three dimensions: regulation at the host cell level, interference of the viral life cycle, and regulation of innate immunity. The specific details are as follows: (i) the first step for coronaviruses to invade host cells is the binding of the viral surface spike protein (S protein) to host cell receptors (such as SARS-CoV-2 binding to ACE2 and FIPV binding to feline aminopeptidase N). (ii) UDCA can inhibit viral replication

and assembly, interfere with the viral life cycle. If the virus successfully invades the cell, UDCA can interfere with the viral replication and assembly process by targeting the metabolic pathways of the host cell or directly affecting the function of viral proteins. (iii) Regulate the host's innate immunity and enhance the antiviral response. Coronavirus infection inhibits the host interferon pathway through multiple mechanisms to evade immune clearance (as described in the article, the FIPV N protein antagonizes IFN-β). UDCA can reverse the immune escape of viruses by activating the host's immune signaling pathways. Activating the interferon signaling pathway, UDCA can promote the expression and activation of host cell pattern recognition receptors, enhance their ability to recognize viral nucleic acids, and thereby activate transcription factors such as IRF3/IRF7, promoting the secretion of IFN-β. IFN-β can induce the expression of downstream antiviral genes (such as ISG15 and MX1) and inhibit viral replication.

In this study, the CRFK-PHBLV-N cell line stably expressing N protein was first constructed for infecting FIPV-CQ strain, and subsequently, through the experimental design of different dosing sequences (pre-treatment, co-treatment, and post-treatment), the effect of UDCA anti-FIPV co-treatment group was verified to be significant by Western blot, RT-qPCR, and IFA assays, which indicated that UDCA may have a direct inactivating effect on FIPV, as well as an effect on the replication stage after the virus enters the host cell. In order to confirm which stage UDCA acted against FIPV, we verified the effect of UDCA by administering it to the viruses at the stages of direct inactivation, adsorption, internalization, replication, and release, and the results of Western blot and RT-qPCR showed that UDCA mainly acted at the stages of direct inactivation, internalization, and replication of the viruses, but the direct inactivation was the most significant effect. The concentrated virus liquid was subjected to phosphotungsten-negative staining transmission electron microscopy, and the results showed that UDCA had a destructive effect on the virus particles, which could destroy their structural integrity and lead to the efflux of their inner materials.

Transcriptomic analysis showed that differentially expressed genes were mainly enriched in IL-17 signaling pathway, JAK-STAT signaling pathway, natural killer cell-mediated cytotoxic effects, and VEGF signaling pathway after FIPV infection. These results suggest that FIPV infection significantly alters the defense responses and metabolic processes of host cells. Further studies revealed that both FIPV infection and UDCA treatment were closely related to the JAK-STAT signaling pathway (Fig. 9). UDCA activated TLR3 to stimulate IRF3 secretion and transcriptionally enhanced the secretion of host antiviral INF-β, which activated the JAK-STAT signaling pathway. The upstream IFN-β and IRF3, midstream STAT1, and downstream ISG15 and VEGF signaling pathways were also activated. The effector molecules, such as ISG15 and MX1, in the midstream and downstream of this pathway, were significantly changed. The experiment confirmed that UDCA treatment of the CRFK-PHBLV-N infection model resulted in a significant increase in the level of IFN-β secretion, enhanced STAT1 phosphorylation and nuclear translocation, as well as a significant upregulation of the transcription and protein expression levels of effector molecules, such as ISG15 and MX1. These results suggest that UDCA can exert antiviral effects by activating the JAK-STAT signaling pathway. This finding is consistent with recent findings on the regulation of the interferon pathway by bile acids (13, 14). It has been shown that inhibition of FXR can deregulate its negative regulation of IRF3 transcriptional activity, thereby enhancing β-interferon production (15, 16). UDCA in the present study may antagonize the inhibitory effect of FXR on IRF3 through a similar molecular mechanism (13, 17), promoting IFN-β secretion. Notably, FIPV infection exhibited the opposite effect to UDCA, being able to significantly inhibit IFN-β secretion and STAT1 activation, leading to downregulation of ISGs expression. This phenomenon is consistent with the known immune escape strategy of coronaviruses.

Several studies have shown that coronaviruses can interfere with the host interferon response through multiple mechanisms (18). For example, the S protein of PEDV inhibits interferon production by interacting with the epidermal growth factor receptor to activate downstream signaling pathways (19); its N protein not only inhibits NF-κB

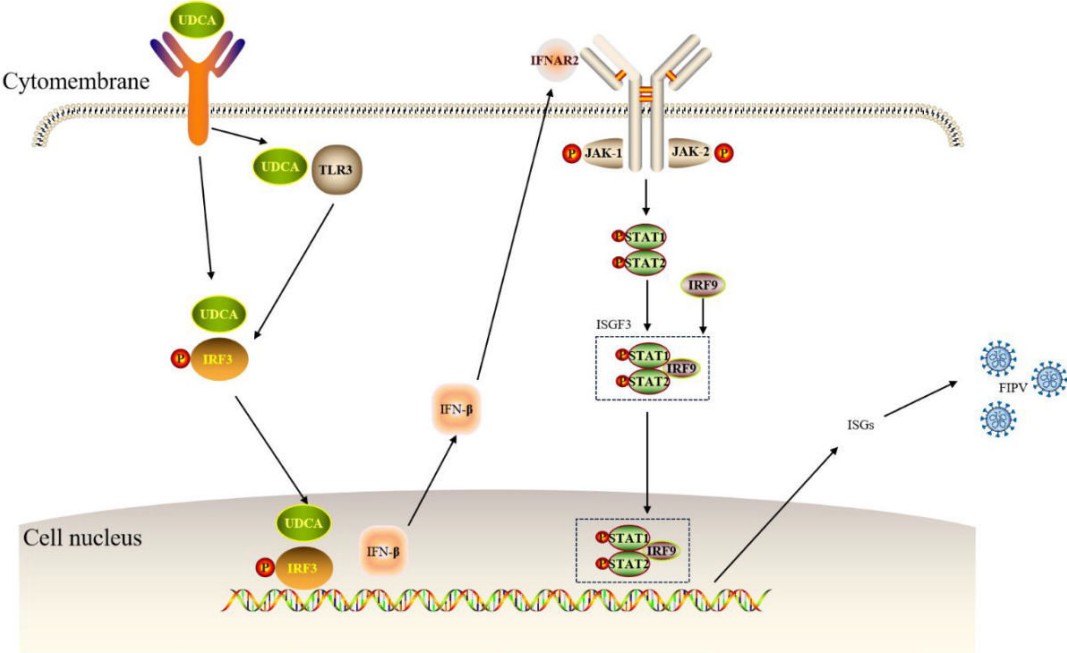

**FIG 9** Diagram of UDCA anti-FIPV mechanism through JAK-STAT signaling pathway.

nuclear translocation but also directly binds to TANK-binding kinase 1 (TBK1) (20), blocking the phosphorylation of IRF3 and its nuclear translocation; and the E protein inhibits the nuclear translocation of IRF3 through direct interaction with IRF3. E protein inhibits nuclear translocation by interacting directly with IRF3. In addition, the non-structural protein Nsp1 interferes with IFN production by degrading CREB-binding protein. It was also found that Nsp1 from seven α-coronaviruses, including FIPV, significantly inhibited the phosphorylation of the STAT1-S727 site (21) and interfered with type I interferon signaling. Other viruses, such as porcine reproductive and respiratory syndrome virus and PDCoV, also interfere with the signaling pathway through different mechanisms (22, 23). These findings suggest that FIPV can inhibit the host's natural immune response through similar molecular mechanisms.

In conclusion, in this study, the efficacy and mechanism of UDCA against FIPV infection were investigated by constructing an *in vitro* FIPV infection model through various cellular experiments combined with transcriptomic analysis, and the results of the study elucidated the antiviral mechanism of UDCA (Fig. 9). Specifically, UDCA can directly destroy the structure of FIPV virus, inhibit the JAK-STAT signaling pathway, block viral invasion, and enhance the host immune response to FIPV infection. This study supports that UDCA can be a potential drug candidate for FIPV prevention, filling the current gap in anti-FIPV small molecule drug research with a new target of drug action. It has great significance for improving the current status of FIP clinical treatment.

## ACKNOWLEDGMENTS

The authors gratefully acknowledge Zhenhui Song, Xingcui Zhang, Zhiwei Sun, and other veterinary medicine students from Southwest University for valuable suggestions and assistance.

This work was supported by the Fundamental Research Funds for the Central Universities (grant number: XDJK2020RC001), the Venture & Innovation Support Program for Chongqing Overseas Returnees 327 (grant number: cx2019097), and the National Natural Science Foundation of China (grant number: 32573339).

Zhenhui Song and Xingcui Zhang conceived the study. Yi Zhong participated in the design of the study, drafted the manuscript, and conducted the experiments. Zhiwei

Sun participated in the design of the study and conducted the experiments. Ziyan Song assisted with the statistical analysis. Jinman Ding, Yanwen Song, Yi Li, Guisong Liao, Xin Wang, Yan Zen, and Nan Hu participated in the study. All authors read and approved the final manuscript.

## AUTHOR AFFILIATION

[1]College of Veterinary Medicine, Southwest University, Chongqing, China

## AUTHOR ORCIDs

Xingcui Zhang http://orcid.org/0009-0003-8551-6910

## FUNDING

| Funder | Grant(s) | Author(s) |
| --- | --- | --- |
| Fundamental Research Funds for Central Universities of the Central South University | XDJK2020RC001 | Zhenhui Song |
| Venture and Innovation Support Program for Chongqing Overseas Returnees | cx2019097 | Zhenhui Song |
| National Natural Science Foundation of China | 32573339 | Zhenhui Song |

## AUTHOR CONTRIBUTIONS

Yi Zhong, Conceptualization, Data curation, Formal analysis, Investigation, Methodology, Software, Validation, Visualization | Zhiwei Sun, Investigation, Methodology, Validation | Ziyan Song, Validation, Visualization, Writing – original draft | Jinman Ding, Writing – original draft | Yanwen Song, Writing – review and editing | Yi Li, Writing – review and editing | Guisong Liao, Writing – review and editing | Xin Wang, Software | Yan Zeng, Software | Nan Hu, Software | Xingcui Zhang, Project administration, Supervision | Zhenhui Song, Funding acquisition, Project administration, Resources

## DATA AVAILABILITY

All data underlying the results are available in the article, and no additional source data are needed.

## ADDITIONAL FILES

The following material is available online.

Open Peer Review

**PEER REVIEW HISTORY (review-history.pdf).** An accounting of the reviewer comments and feedback.

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
