## [Reviewer comments · Microbiology Spectrum]

Microbiology Spectrum

Ursodexychoic Acid Inhibits Feline Infectious Peritonitis Virus Infection through Activating JAK-STAT Signaling Pathway Induced Type I Interferon

yi zhong, Zhi Sun, Ziyang Song, jinman ding, yanwen song, Yi Li, Gui Liao, Xingcui Zhang, and Zhenhui Song

Corresponding Author(s): Zhenhui Song, Southwest University

Review Timeline:

Submission Date:	November 10, 2025
Editorial Decision:	December 1, 2025
Revision Received:	January 6, 2026
Accepted:	February 3, 2026

Editor: Yunyu Chen

Reviewer(s): Disclosure of reviewer identity is with reference to reviewer comments included in decision letter(s). The following individuals involved in review of your submission have agreed to reveal their identity: Tongling Shan (Reviewer #1); Bin Li (Reviewer #2)

Transaction Report:

DOI: <https://doi.org/10.1128/spectrum.03633-25>

Re: Spectrum03633-25 (Ursodeoxycholic Acid inhibiting Feline Infectious Peritonitis Virus infection through activating JAK-STAT signaling pathway induced type I interferon)

Dear Dr. Song Zhenhui:

Thank you for the privilege of reviewing your work. Your manuscript has now been reviewed by expert reviewers for the journal. The reviewers have raised some important issues and have made many useful suggestions for its improvement. Based on these comments (attached below), as well as my own review, we would be happy to receive a revised version of the manuscript that is responsive to the reviewers' comments.

Revision Guidelines

Sincerely,
Yunyu Chen
Editor
Microbiology Spectrum

Reviewer #1 (Comments for the Author):

This manuscript systematically evaluate the antiviral activity of ursodeoxycholic acid (UDCA) against Feline infectious peritonitis virus (FIPV). The in vitro assays demonstrated that UDCA extracts significantly inhibit FIPV through directly disrupt the viral envelope components, inducing the disintegration of the viral structure; UDCA can also significantly promote the secretion of IFN- β in CRFK cells, enhance the phosphorylation and nuclear translocation of STAT1, and upregulate the expression of

interferon-stimulated genes ISG15 and MX1 by activating the JAK-STAT signaling pathway. These findings provide important theoretical basis for developing antiviral drugs targeting FIPV. The hypothesis is supported by the majority of the data. The authors need to address these issues to strengthen their conclusions and improve the overall quality of the manuscript.

1. In the Discussion section, please elaborate on the mechanism of action of UDCA against the coronaviruses included in the study (e.g. SARS-CoV-2, TGEV).
2. Line 364, FXR should be replaced with RIPV.
3. Please specify the exact viral titer used in the virus infection cell experiments in the Methods section.
4. The study results indicate that UDCA can significantly inhibit the internalization of FIPV. Why didn't the authors further elucidate this mechanism?
5. Line 224 is incorrect; Fig. 5E should be Fig. 5D.
6. To support the research conclusions, the authors need to further validate the preventive and therapeutic effects of UDCA against FIPV at the animal level.
7. Please specify in the text how many-fold the viral titer can be increased by constructing a stable cell line expressing FIPV N.
8. The writing of the article needs to be further improved.

Reviewer #2 (Comments for the Author):

This study demonstrated the efficacy and mechanism of the UDCA against FIPV infection were investigated by constructing an in vitro FIPV infection model through various cellular experiments combining with transcriptomic analysis, and the results of the study elucidated the antiviral mechanism of UDCA. Specifically, UDCA can directly destroy the structure of FIPV virus, inhibit the JAK-STAT signaling pathway, block viral invasion, and enhance the host immune response to FIPV infection. This is an interesting idea for understanding the mechanism of the UDCA against FIPV infection, and and this study is also important for the discovery of new targets for the prevention and treatment of FIPV infection. There are some areas that should be modified accordingly to clarify and add scientific significance to the manuscript.

1. Figure 4B shows that UDCA treatment during both the viral internalization and replication stages can significantly reduce the expression level of FIPV-N protein. Please analyze the possible mechanisms by which UDCA inhibits viral internalization in the discussion section of the manuscript.
2. Line 224: Fig. 5E should be Fig. 5D.
3. In the experiments demonstrating that stabilization of cells enhances FPIV infection titers. How many-fold the virus titer can be increased through the construction of a cell line stably expressing FIPV N.
4. In Fig. 5, correlation analysis of FIPV-UDCA co-treatment with antiviral gene expression. Why is there no validation result for IRF3?
5. There are some problems with the use of the English language and wording throughout the manuscript. It would greatly benefit from editing by native English speaker.

Dear reviewers,

We are very grateful to your comments for the manuscript. According to your advice, we have amended the relevant part in manuscript. Your questions were answered below. (Note: All the lines and pages are from our revised manuscript with revision marks)

Reviewer 1

Comment 1: In the Discussion section, please elaborate on the mechanism of action of UDCA against the coronaviruses included in the study (e.g. SARS-CoV-2, TGEV).

Response 1: Thank you for your suggestion. We have elaborated on the mechanism of action of UDCA against the coronaviruses included in the study. (Line 538-556)

Comment 2: Line 364, FXR should be replaced with RIPV.

Response 2: Thank you for your suggestion. The “FXR” is the abbreviation of farnesol X receptor, isn’t a mistake in writing of “FIPV”.

Comment 3: Please specify the exact viral titer used in the virus infection cell experiments in the Methods section.

Response 3: Thank you for your suggestion. We used the FIPV for infection with MOI=1 (Line 173).

Comment 4: The study results indicate that UDCA can significantly inhibit the internalization of FIPV. Why didn't the authors further elucidate this mechanism?

Response 4: This study mainly focused on the inhibitory effects and mechanisms of the replication phase, and we will further carry out the effects of UDCA on internalization.

Comment 5: Line 346 is incorrect; Fig. 5E should be Fig. 5D.

Response 5: Thank you for your question. We have changed the Fig. 5E to Fig. 5D (Line 412).

Comment 6: From Figure 1G, we can observe that the expression of the N protein in

the stable cell line shows a decreasing trend from the first generation to the thirtieth generation. Does this indicate that the stability of this cell line is not strong? Did it have any impact on your experiment?

Response 6: Thank you for your good questions. It should be noted that, with the increase of the number of passages, the cells gradually enter into the state of senescence, which is manifested by the decrease of metabolic activity, and then may affect the basic biological processes of gene transcription and translation, and ultimately leads to the decrease in the expression level of the target protein. Based on the findings, in order to ensure the reliability and consistency of the results, in this study, we selected cells with no more than 10 passages for the experiments.

Comment 7: Please specify in the text how many-fold the viral titer can be increased by constructing a stable cell line expressing FIPV N.

Response 7: Thank you for your suggestion. We have already added the relevant content to the manuscript (Line 283-287).

Comment 8: The writing of the article needs to be further improved.

Response 8: Thank you for your suggestion. We have revised the entire article.

Reviewer 2

Comment 1: Figure 4B shows that UDCA treatment during both the viral internalization and replication stages can significantly reduce the expression level of FIPV-N protein. Please analyze the possible mechanisms by which UDCA inhibits viral internalization in the discussion section of the manuscript.

Response 1: Thank you for your suggestion. We have elaborated on the mechanism of action of UDCA against the coronaviruses included in the study. (Line 538-556)

Comment 2: Line 346: Fig. 5E should be Fig. 5D.

Response 2: Thank you for your question. We have changed the Fig. 5E to Fig. 5D (Line 412).

Comment 3: In the experiments demonstrating that stabilization of cells enhances FPIV infection titers. How many-fold the virus titer can be increased through the construction of a cell line stably expressing FIPV N.

Response 3: Thank you for your suggestion. We have already added the relevant content to the manuscript. (Line 283-287).

Comment 4: In Figure 1H, the DAPI and Merge groups in the CRFK-PHBLV-N-Vector and CRFK-PHBLV- groups are reversed.

Response 4: Thank you for your kind remind. We have made modifications to Figure 1H.

Comment 5: There are some problems with the use of the English language and wording throughout the manuscript. It would greatly benefit from editing by native English speaker.

Response 5: Thank you for your suggestion. We have revised the entire article.

Re: Spectrum03633-25R1 (Ursodexychoic Acid Inhibits Feline Infectious Peritonitis Virus Infection through Activating JAK-STAT Signaling Pathway Induced Type I Interferon)

Dear Dr. Zhenhui Song:

Your manuscript has now been reviewed by expert reviewers for the journal. Based on the reviewer's comments, as well as my own review, I would be happy to accept your manuscript in its current form for publication in Microbiology Spectrum.

Your paper will first be checked to make sure all elements meet the technical requirements. ASM staff will contact you if anything needs to be revised before copyediting and production can begin. Otherwise, you will be notified when your proofs are ready to be viewed.

Sincerely,
Yunyu Chen
Editor
Microbiology Spectrum

Reviewer #2 (Comments for the Author):

All issues have been addressed.